# Survival After Shunt Therapy in Normal-Pressure Hydrocephalus: A Meta-Analysis of 1614 Patients

**DOI:** 10.3390/neurolint16060107

**Published:** 2024-11-11

**Authors:** Johannes Wach, Agi Güresir, Erdem Güresir, Martin Vychopen

**Affiliations:** Department of Neurosurgery, University Hospital Leipzig, 04103 Leipzig, Germany; johannes.wach@medizin.uni-leipzig.de (J.W.); agi.gueresir@medizin.uni-leipzig.de (A.G.); erdem.gueresir@medizin.uni.leipzig.de (E.G.)

**Keywords:** baseline symptom burden, meta-analysis, normal-pressure hydrocephalus, survival analysis, ventriculoperitoneal shunt therapy

## Abstract

Background: Ventriculoperitoneal (VP) shunt therapy is a crucial intervention for normal-pressure hydrocephalus (NPH). This meta-analysis delves into survival time and the impact of baseline symptom burden on survival after VP shunt therapy for NPH, employing reconstructed pooled survival curves and a one-stage meta-analysis. Methods: IPD regarding overall survival (OS) were acquired from published Kaplan–Meier charts, utilizing the R package IPDfromKM in R (Version 4.3.1, the R Foundation for Statistical Computing). Reconstructed Kaplan–Meier charts were then generated from the pooled IPD data. Both one-stage and two-stage meta-analyses were executed, with hazard ratios (HRs) employed as metrics to evaluate effectiveness. Results: From the initial screening of 216 records, five articles encompassing 1614 patients met the eligibility criteria for inclusion. In two of the five included studies, overall survival was stratified by gait score (1–4 vs. ≥4) in 1043 patients, continence score (1–3 vs. ≥4) in 1022 patients, and mRS (0–2 vs. ≥3) in 956 patients. Patients with good gait demonstrated a mean survival of 8.24 years, while those with poor gait had a mean survival of 6.19 years (log-rank test: *p* < 0.001). The HR for gait was 2.25 (95% CI: 1.81–2.81, *p* < 0.001). Continence score stratification revealed a significant difference in survival time (log-rank test: *p* < 0.001), with an HR of 1.66 (95% CI: 1.33–2.06, *p* < 0.001). Similarly, mRS stratification demonstrated a significant survival difference (log-rank test: *p* < 0.001), with an HR of 2.21 (95% CI: 1. 74–2.80, *p* < 0.001). The reconstructed survival curves for all NPH patients treated with VP shunt therapy, pooling data from five studies, revealed a median survival time of 8.82 years (95% CI: 8.23–9.40). Survival rates at 1, 3, 5, 7, 9, 11, and 13 years were 95.7%, 83.8%, 70.5%, 59.5%, 48.7%, 35.8%, and 25.4%, respectively. Comparison with a general control population showed an HR of 1.79 (95% CI: 1.62–1.98, *p* < 0.001). Conclusions: This comprehensive meta-analysis underscores the influence of baseline symptom burden on survival after VP shunt therapy in NPH. Therapy in the early stages for those without significant comorbidities may enhance survival.

## 1. Introduction

Normal-pressure hydrocephalus (NPH) is a prevalent condition that significantly affects daily activities and quality of life [1]. Its incidence among individuals older than 65 years is reported to be 3.7%, with expectations of a rise in tandem with the aging population [2]. Recent findings by Peterella et al. [3] suggest that these reported rates may substantially underestimate the true incidence within the population. The natural history of NPH patients is characterized by a poor prognosis, markedly impairing quality of life and potentially reducing life expectancy [4].

The clinical symptoms of this chronic disease, known as the “Hakim” triad, include gait disturbance, bladder incontinence, and dementia [5]. Left untreated, the long-term progression of the disease can severely impair the well-being of the patient. Currently, the optimal therapy for NPH typically involves invasive diagnostics such as lumbar puncture or lumbar drainage [6], which aim to simulate the artificial drainage of the cerebrospinal fluid (CSF) and confirm the suspicion of NPH [7]. If there is a positive clinical response to the diagnostic procedures, definitive surgical treatment via ventriculoperitoneal (VP) shunt surgery is recommended.

However, the precise magnitude of the long-term clinical benefits, as assessed by longitudinal data from patients undergoing surgery for normal-pressure hydrocephalus, remain the subject of intensive investigation [8,9]. Given the chronic nature of this condition, effective symptom management may lead to improvements in quality of life, and in the case of gait disturbance/ataxia control, may even enhance overall functional performance and potentially extend survival time [10,11].

While numerous retrospective single-center observational cohort studies have investigated long-term outcomes, an individual patient data (IPD) meta-analysis has yet to be conducted. The objective of this study is to assess the impact of CSF shunt implantation on the long-term survival outcomes of patients with NPH.

## 2. Materials and Methods

This systematic review and meta-analysis adhered to the guidelines outlined in the Preferred Reporting Items for Systematic Reviews and Meta-Analyses (PRISMA) statement, specifically tailored for IPD development cohorts. The study protocol was prospectively registered in the “International Prospective Register of Systematic Reviews” (PROSPERO) in 2024 [12]. The detailed, pre-specified protocol is accessible upon request. The PRISMA checklist is available in the Appendix A.

### 2.1. Search Strategy and Study Inclusion

We conducted a literature search in the PubMed, Medline, Cochrane and Embase databases for all clinical studies regarding long-term outcome of normal-pressure hydrocephalus treatment. Studies published in English were retrieved. The search strategy was conducted based on the PICOS criteria [13].

The following mesh terms were used to identify eligible studies: (1) “normal pressure hydrocephalus” AND “survival”; (2) “normal pressure hydrocephalus” AND “mortality”. Inclusion criteria required long-term survival follow-up data displayed in Kaplan–Meier charts with corresponding number-at-risk tables for reconstruction of IPD. The control group data were extracted from the study by Andrén et al. [14], representing a representative cohort from the normal population.

Included were all studies reporting on OS and long-term follow-up data of patients with normal-pressure hydrocephalus treated by shunt insertion. Two reviewers (J.W., M.V.) independently screened abstracts and full-text articles for two rounds, with any residual conflicts resolved by a supervising third reviewer (E.G.).

### 2.2. Quality Assessment

The assessment of quality and risk of bias in the included studies was conducted using the National Institutes of Health Quality Assessment Tool for observational cohort and cross-sectional studies (NIH-QAT) [15].

### 2.3. Data Extraction

Two reviewers (M.V., J.W.) independently extracted the following characteristics from the publications: year of publication, country, study timeframe, total number of patients, age, sex, NPH patients’ symptom gradings (ordinal grading systems for the following symptoms: gait, continence, and modified Rankin scale (mRS)), and reported median survival time [16,17]. The individual patient data for survival analysis were obtained by digitizing the published Kaplan–Meier survival curves and number-at-risk tables using DigitizeIt (Version 2.5.10 for macOS, Braunschweig, Germany) [18].

The extracted information on survival and the published number-at-risk tables were utilized to reconstruct the Kaplan–Meier curves for each included study, following the method outlined by Liu et al. [19] and implemented using the R package IPDfromKM in R (version 4.3.1, R Foundation for Statistical Computing, Vienna, Austria). Additionally, number-at-risk tables were generated. Comparisons stratified by symptom burden (measured by gait, continence, and mRS) were performed between the reconstructed curves, risk tables, estimated hazard ratios (HRs), and estimated 95% confidence intervals (CIs) with the corresponding data in the original publications. In cases of apparent discrepancies, the information extraction process was repeated.

### 2.4. Statistics

The IPD information of all survival time data from all the included studies was pooled, and Kaplan–Meier charts of OS were created using the R packages Survminer and Survival in R software version 4.3.1 (R Foundation for Statistical Computing, Vienna, Austria). The 1, 3-, 5-, 7-, 9-, 11-, and 13-year survival rates were constructed. The hazard ratios of each included investigation as well as the pooled HR and corresponding 95% CI between patients with continuous Lev treatment and those without were calculated. In the two-stage meta-analysis, the estimated HRs and the corresponding 95% CIs of each study were combined using both fixed-effects and random-effects models with the generic inverse variance method. 

The hazard ratios were estimated and then transformed into natural logarithms (LNs). The standard errors (SEs) for each study were computed from the 95% confidence interval (CI) using the formula SE = (LN (upper CI limit) − LN (lower CI limit))/3.92, in accordance with the guidelines provided in the Cochrane Handbook for Systematic Reviews of Interventions, version 6.420. The significance of each study’s relative contribution, determined by its sample size, was considered when estimating treatment effects in a random-effects model. The combined estimates were visually presented in forest plots using the Review Manager Web (RevMan Web version 5.4.1 from The Cochrane Collaboration. Statistical significance was defined as *p* < 0.05. To assess publication bias, funnel plots were employed and visually examined [20].

## 3. Results

### 3.1. Study Selection and Study Characteristics

We identified 216 studies eligible for screening. Following title screening and abstract screening, 51 studies remained for full-text review. Exclusion criteria encompassed studies involving pediatric patients, those lacking reports on long-term outcome or those that did not report on number-at-risk tables regarding survival probabilities, technical notes, case studies, and studies focusing on other types of hydrocephalus. Subsequently, after thorough review of 51 manuscripts, 5 articles with long-term longitudinal survival data displayed in Kaplan–Meier curves with number-at-risk tables were included in the longitudinal IPD meta-analysis of overall survival, while 2 of these five studies were included in longitudinal survival data analysis stratified by gait score, continence score and modified Rankin scale (mRS) (see Figure 1 displaying the PRISMA flow diagram).

The characteristics of the included studies are presented in Table 1.

The present meta-analysis comprises data from 1043 patients for the analysis of baseline symptoms and 1614 patients for the overall survival analysis. Andrén et al. [14] conducted a study with a large cohort of NPH patients, alongside a sizable control cohort representative of the normal population. Moreover, this study extensively reports on baseline symptoms, including gait score, continence score, and mRS.

In a separate study by Andrén et al. [21], a comparison cohort between early and delayed shunt implantation in NPH patients at a single institution is presented. This retrospective study identified delays in shunt implantation primarily attributable to administrative and economic issues, resulting in unexpected postponement of elective procedures.

The investigation by Eklund et al. [22] offers a long-term population-matched control study of NPH patients. Eklund et al. [22] provides insights into the overall survival of both groups, as well as the causes of death and comorbidities.

Furthermore, Pyykkö et al. [9] conducted a single-center retrospective investigation of NPH patients treated in Finland, compared to a case-matched control group. Additionally, they report on comorbidities and the results of frontal brain biopsies. 

Klinge et al. [23] presents a single-surgeon retrospective cohort study of NPH patients. The study delves into overall survival, gait score, continence score, Charlson comorbidity index, and mRS.

All five studies reported Kaplan–Meier charts and number-at-risk tables regarding the overall survival of patients undergoing shunt implantation because of NPH.

### 3.2. Reconstructed Pooled Survival Curves and One-Stage Meta-Analysis of the Impact of Baseline Symptom Burden on Survival After VP Shunt Therapy for NPH

Two articles encompassing 1043 patients were deemed eligible for inclusion in this analysis. IPD reconstruction of survival times by gait score (1–4/≥5), continence score (1–3/≥4), and mRS (0–2/≥3) was performed. OS curves were reconstructed, and side-by-side comparisons were conducted with the original plots. The estimated HRs and corresponding 95% CI of included studies in the one-stage analysis are displayed in Table 2.

The Kaplan–Meier plots reconstructed for each study exhibited notable similarity when compared to the published plots, and any differences in the number-at-risk tables were minimal. The median (IQR) follow-up time for OS of the reconstructed IPD regarding gait score was 4.2 years (2.8–5.8). Figure 2A shows the reconstructed OS curve for the reconstructed cohort stratified by baseline gait score (1–4/≥5). The mean survival time of patients with good gait was 8.24 years (95% CI: 7.86–8.62) and in those with poor gait was 6.19 years (95% CI: 5.82–6.55) (log-rank test: *p* < 0.001). The hazard ratio of the pooled IPD for gait was 2.25 (95% CI: 1.81–2.81, *p* < 0.001). Figure 2B illustrates the reconstructed OS curve of the cohort, categorized by continence score (1–3/≥4). Patients with a favorable continence score (1–3) exhibited a mean survival time of 7.06 years (95% CI: 6.80–7.32), while those with an impaired continence score (≥4) had a mean survival time of 6.57 (95% CI: 6.20–6.95) (log-rank test: *p* < 0.001). The hazard ratio from the combined IPD for continence was 1.66 (95% CI: 1.33–2.06, *p* < 0.001). Figure 2C depicts the reconstructed OS curve of the cohort, stratified by mRS (0–2/≥3). Patients with a favorable mRS (0–2) demonstrated an average survival of 8.31 years (95% CI: 7.93–8.68), whereas those with a poor mRS (≥3) had a mean OS of 6.08 years (95% CI: 7.14–7.74) (log-rank test: *p* < 0.001). The hazard ratio derived from the pooled IPD for mRS was 2.21 (95% CI: 1.74–2.80. *p* < 0.001).

### 3.3. Reconstructed Pooled Survival Curves of All NPH Patients Treated by VP Shunt Therapy

The reconstructed survival times from five studies [9,14,21,22,23] were further pooled in an analysis of all available probabilities of survival in NPH. Figure 3A illustrates the reconstructed OS curve for the total cohort, with 1614 patients. The median survival time in NPH patients treated with a VP shunt was 8.82 years (95% CI: 8.23–9.40). The 1, 3-, 5-, 7-, 9-, 11-, and 13-year survival rates after VP shunt therapy for NPH were 95.7%, 83.8%, 70.5%, 59.5%, 48.7%, 35.8%, and 25.4%. The total reconstructed IPD was also compared to a general control population from the study by Andren et al. [14] (see Figure 3B). The hazard ratio for NPH patients compared to the control was 1.79 (95% CI: 1.62–1.98, *p* < 0.001).

### 3.4. Two-Stage Meta-Analysis of Overall Survival in Normal-Pressure Hydrocephalus Patients

To ensure the reliability of the results, a two-stage meta-analysis was undertaken. In terms of overall survival, random-effects models with the generic inverse variance analysis of the effect measures (hazard ratio) were performed for the studies by Andren et al. [14] and Klinge et al. [23]. In terms of overall survival stratified by gait score (1–4/≥5), continence score (1–3/≥4), and mRS (0–2/≥3) the pooled hazard ratios were 2.20 (95% CI: 1.33–3.63), 1.59 (95% CI: 1.06–2.39), and 2.19 (95% CI: 1.28–3.75), respectively (see Figure 4a–c). This affirms an association between the baseline symptom burden and survival time in VP shunt-treated NPH patients. The evaluation revealed no significant heterogeneities among the analyzed factors.

### 3.5. Bias and Quality Evaluation

To comprehensively evaluate the potential for publication bias, a multifaceted approach was employed. This began with a rigorous and comprehensive literature search to ensure all relevant studies were identified. Subsequently, only studies that met strict, predefined inclusion and exclusion criteria were selected for the meta-analysis, ensuring consistency and reliability in the data used. Finally, the evaluation of publication bias was carried out by scrutinizing funnel plots related to the survival analysis for each analyzed factor (refer to Figure 5A–C).

The quality assessment was conducted using the NIH-QAT tool, which yielded favorable ratings for the studies included. Detailed ratings for each of the 14 NIH-QAT domains are available in Appendix A. It should be noted that none of the studies provided information on the level of exposure regarding shunt valve programming. Additionally, due to the retrospective nature of the studies, blinding of patients and physicians was not feasible.

## 4. Discussion

We conducted an IPD meta-analysis of overall survival in a cumulative cohort of 1614 patients, juxtaposed with a control of 4890 patients, as reported by Andrén et al. [14]. The median survival of NPH patients was notably shorter compared to the general population. Our longitudinal analysis, stratified by the severity of the baseline symptoms (gait score ≥ 5, continence score ≥ 4, and mRS ≥ 3) at treatment initiation, demonstrated significantly reduced survival time in patients with a higher burden of NPH symptoms. This finding was corroborated by a two-stage meta-analysis. Appendix A summarizes the results.

### 4.1. Shunt-Related Deaths—Infection and Obstruction

While infection and obstruction are feared complications of VP shunt implantation, the prospective multicentric Japanese study by Hashimoto et al. [24] reported that only 1% of the 100 NPH patients underwent revision surgery due to shunt catheter obstruction. 

Conversely, subdural hematomas in overdrained shunted NPH patients can pose challenges, requiring management of both the hematoma and the shunt’s drainage function, thereby regulating the symptom burden of NPH. Eklund et al. [22] demonstrated a significantly higher probability of NPH patients dying from subdural hematoma compared to the general population, indicating a surgery-dependent long-term complication due to overdrainage and subsequent subdural hematoma development (SDH) [25]. The risk of SDH development in NPH patients has been reported to be approximately 30%, rendering it a serious and potentially fatal complication of NPH treatment [26]. To mitigate this risk, it is advisable to utilize a programmable valve equipped with standard pressure controls to minimize the likelihood of this complication [27,28]. In cases where antiplatelet or anticoagulation therapy is necessary, the risk of SDH development may be further increased and exacerbated by overdrainage [29].

### 4.2. Pathophysiology and Risk Factors for Clinical Deterioration 

Eklund et al. [22] demonstrated a very high inverse correlation between the survival time of NPH patients and the severity of cardiovascular comorbidities. The strong correlation suggests a vascular pathophysiological mechanism underlying NPH, potentially leading to a subsequent decrease in life expectancy among NPH patients as a result of metabolic, neurovascular, and cardiovascular-related conditions (e.g., hypertension, stroke, diabetes mellitus, and white-matter lesions) [30,31]. Emerging evidence debates the role of the glymphatic system in NPH. For the glymphatic system to function properly, it relies on intact aquaporin 4 channels, appropriate arterial pulsation, and healthy sleep patterns [32]. In individuals with NPH, the glymphatic system is compromised, leading to a diminished ability to clear neurotoxic substances such as beta amyloid and hyperphosphorylated tau [33]. This disruption in the glymphatic system can lead to the accumulation of harmful substances, potentially causing astrogliosis, neuroinflammation, and neuronal dysfunction [34]. Furthermore, several studies suggest that cerebral blood flow and regional cerebral blood flow, as assessed by imaging modalities such as single-photon emission computed tomography, are compromised in NPH patients [35,36,37]. Novel imaging methods such as arterial spin-labeling perfusion can map brain perfusion without contrast agents, and Virhammer et al. [38] demonstrated reduced blood supply in the periventricular white matter, basal ganglia, and thalamus in NPH patients using this technique. The pathophysiological understanding of NPH is not entirely understood. However, the present knowledge is essential because the symptom burden and development of symptoms significantly influence quality of life as well as overall survival time. Furthermore, secondary non-responders of VP shunt therapy for NPH were reported to be found in twenty percent of patients [39]. Hence, there is a need to translate the knowledge about these pathophysiological processes into clinical biomarkers to facilitate the process of identifying patients who would benefit most from VP shunt therapy and also to provide potential future additional drug therapy options. Appendix A summarizes some major pathophysiological pathways facilitating the development and progression of NPH.

### 4.3. Gait Score

The control of gait pattern plays a crucial role in the management of NPH [40]. Falls and traumatic injuries are identified as leading causes of death among NPH patients, as highlighted by Eklund et al. Given that NPH patients constitute a demographic with an inherently elevated risk of falling, Okubo et al. present a randomized control trial demonstrating that maintaining walking abilities is paramount in reducing this risk [41]. As illustrated in Figure 2A, NPH patients with better performance based on gait score exhibit superior overall survival compared to those with severely impaired walking abilities. This underscores the importance of early diagnosis and prompt surgical intervention in NPH management.

### 4.4. Continence Score

The analysis of continence score confirms the theory that patients with severe NPH symptoms may exhibit lower life expectancy compared to those with milder symptoms. This finding further underscores the importance of early diagnosis and timely surgical intervention in NPH management. Given that incontinence significantly impairs daily activities, addressing this symptom promptly may be beneficial in reducing overall mortality and extending lifespan [42].

### 4.5. Clinical Implications

The HRs presented in this study offer critical insights into how baseline symptom severity might influence survival outcomes in patients undergoing VP shunt therapy for NPH. For instance, an HR of 2.25 for gait score (1–4 vs. ≥5) suggests that patients with severe gait impairment face more than twice the risk of mortality during follow-up compared to those with milder impairment. Clinically, this highlights the urgency of timely intervention for patients with moderate gait dysfunction after appropriate diagnostic testing for NPH to optimize survival outcomes. Similarly, the HRs for continence score (1.66) and mRS (2.21) emphasize the strong predictive value of these measures. These findings are consistent with the general evidence linking physical performance metrics, such as gait speed, to survival in elderly patients above 65 years, as shown by Studenski et al. [43], who demonstrated that gait functionally significantly predicted life expectancy across diverse elderly populations in a study of 34,485 patients with data of gait speed. Importantly, urinary incontinence, a cardinal symptom of NPH, is independently associated with increased mortality risk. Damian et al. [44] observed a similar trend in a cohort of 675 nursing home residents, demonstrating that incontinence was associated with a 24% higher risk of mortality, with more severe forms linked to a 44% increase. 

### 4.6. Strength and Limitations

We present an IPD meta-analysis of survival in a large cumulative cohort of NPH patients. All findings support early diagnostic and aggressive surgical treatment to preserve quality of life and extend life expectancy. Furthermore, the relatively low rate of shunt-associated complications in prospective cohorts of NPH patients with programmable valves further justifies the treatment approach [45].

The strong limitation of this study is the retrospective character of the cohorts eligible for meta-analysis. Two of the five included studies are from the same country, which potentially runs the risk of the 102 patients of Andren et al.’s [21] single-center study also being part of the Swedish Hydrocephalus Quality Registry from between 2004 and 2011, with 979 patients [14].

Additionally, the control group was represented by a statistical sample of a “normal population”, which was only available in one study. Due to case-matching controls in other studies and the resulting inhomogeneity of the controls, we decided to include only the homogeneous control group presented in the study published by Andrén et al. 2019. The analysis might have a strong geographical bias, as all the included studies analyze populations from Western Europe/USA, and the preferred shunt system modalities differ among the continents. For instance, in Japan, the most common treatment modality for NPH is the insertion of a lumboperitoneal shunt system, while in Europe and the USA, VP shunt surgery is the treatment of choice [46]. Furthermore, the IPD cannot be further stratified regarding the age at insertion of the VP shunt and the causes of death in each case. 

## 5. Conclusions

Our findings support the proactive diagnostic and therapy of NPH patients in order to preserve their quality of life and prolong their life expectancy. The results highlight the impact of initial symptom severity on survival after VP shunt treatment in NPH patients. Early intervention in individuals with rigorous diagnosed NPH and without substantial comorbidities could potentially enhance their survival time.

## Figures and Tables

**Figure 1 neurolint-16-00107-f001:**
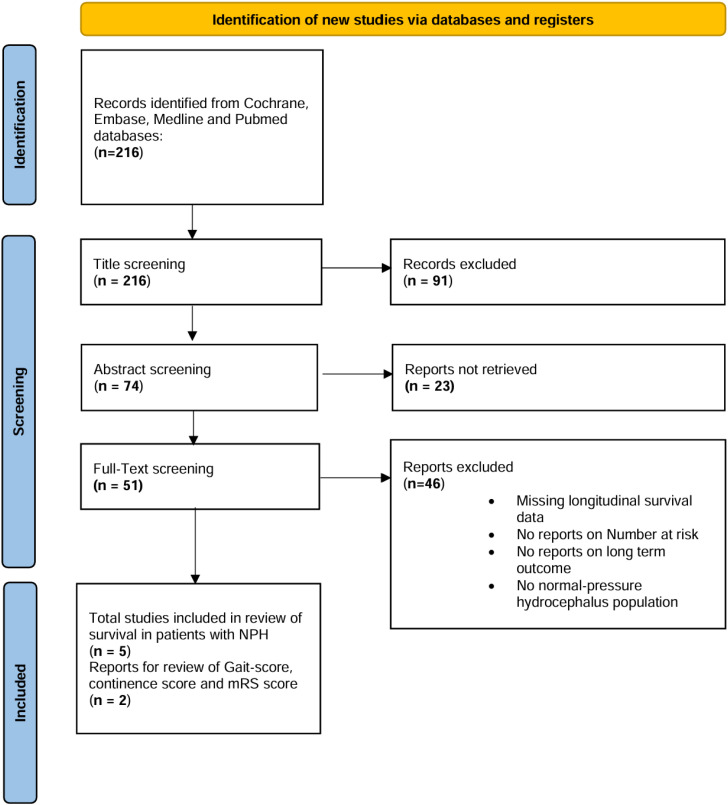
PRISMA flow diagram illustrating the study selection process.

**Figure 2 neurolint-16-00107-f002:**
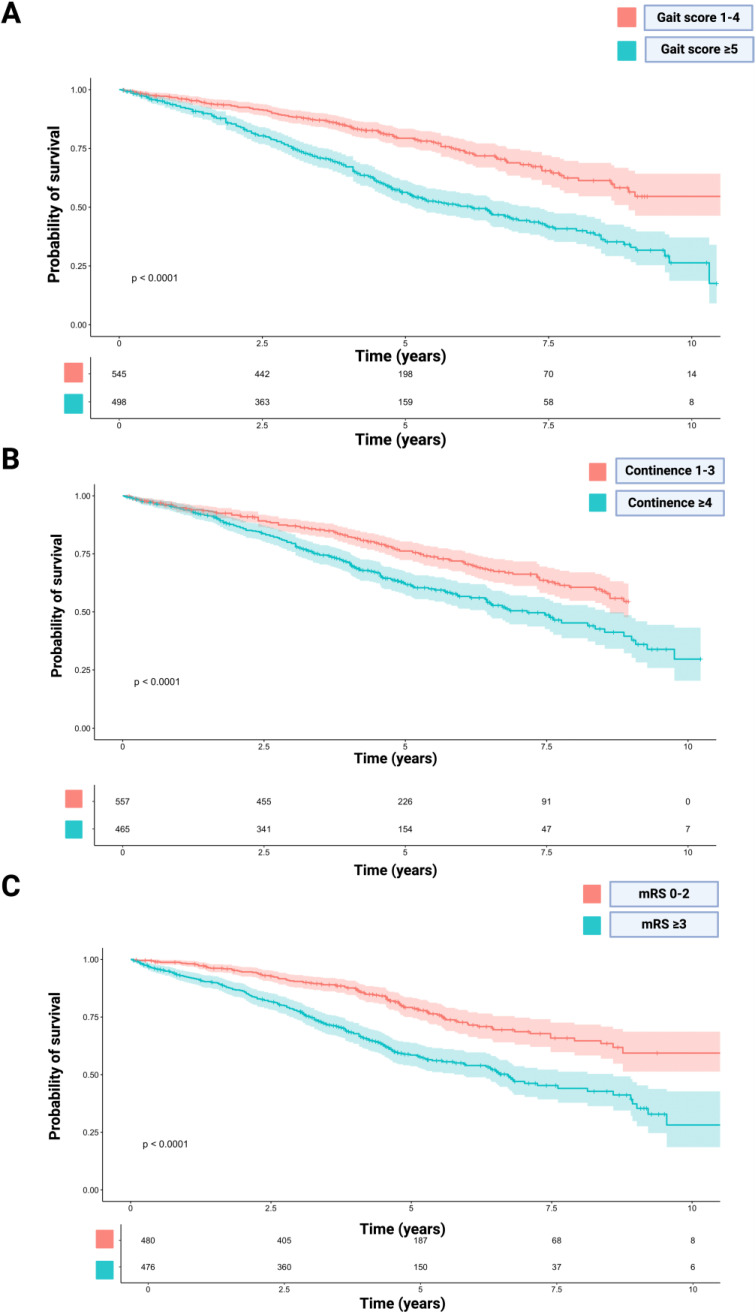
Kaplan–Meier chart displaying survival probability stratified by preoperative gait score (1–4/≥5) (**A**), preoperative continence score (1–3/≥4) (**B**), and preoperative mRS (0–2/≥3) (**C**). The log-rank tests showed a significantly enhanced survival time in patients with a lower burden of NPH symptoms. The shadowed areas surrounding the plots display the confidence intervals. Number-at-risk tables are given below each plot.

**Figure 3 neurolint-16-00107-f003:**
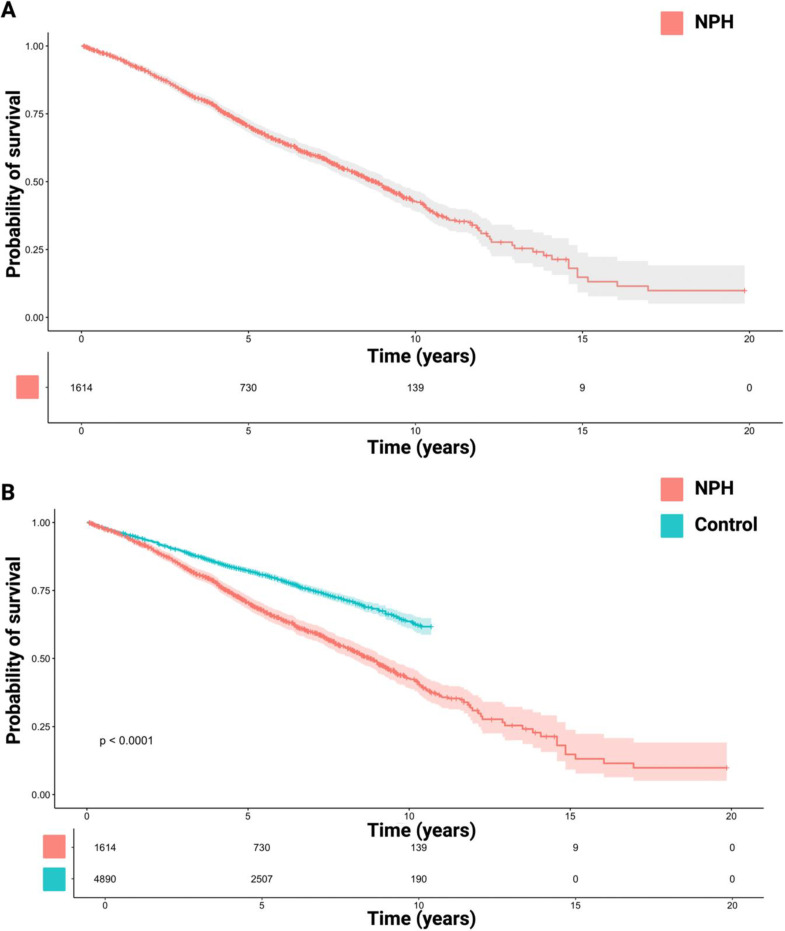
(**A**) Kaplan–Meier chart sowing survival probability in 1614 patients after VP shunt therapy for normal-pressure hydrocephalus. The median survival time in NPH patients treated with a VP shunt was 8.82 years (95% CI: 8.23–9.40). (**B**) The log-rank test (*p* < 0.0001) showed a significantly shorter survival time in patients with surgically treated NPH compared to the general population. The shadowed areas surrounding the plots display the confidence intervals. Number-at-risk tables are given below each plot.

**Figure 4 neurolint-16-00107-f004:**
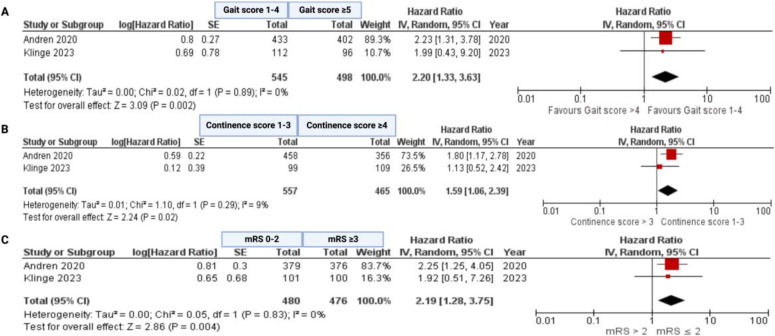
Forest plot displaying log (hazard ratio), log (standard error), HR, and 95% CI estimates for OS in a random-effects model, with the generic inverse variance method of the included studies for analysis of baseline symptom burden in survival of surgically treated NPH patients; Gait score (**A**), Continence score (**B**) and mRS (**C**). *X*-axis locations of squares indicate the hazard ratio. The weight of the included investigations is also reported. The red quadrangles represent the weight of each study. The black diamonds constitute the hazard ratios of the pooled cohort in each model [14,23].

**Figure 5 neurolint-16-00107-f005:**
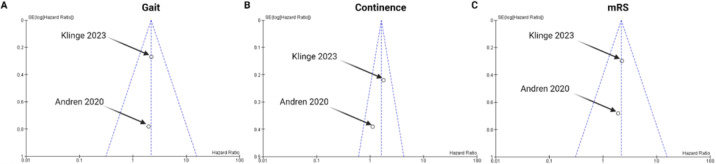
Funnel plot assessment for publication bias from treatment effects of studies included in the survival analysis of baseline symptom burden in surgically treated NPH patients [14,23]; Gait score (**A**), Continence score (**B**) and mRS (**C**).

**Table 1 neurolint-16-00107-t001:** Characteristics of eligible studies.

Study	Year	Country	Reported Timeframe	Study Design/ Treatment Arms	Number of Patients (*n*)	Outcomes
Andrén et al. [14]	2020	Sweden	2004–2011	Retrospective/NPH vs. control group	NPH group = 979 Control group *n* = 4890	Overall survival Gait score Continence score mRS
Andrén et al. [21]	2021	Sweden	2010–2011	Retrospective/delayed vs. early treatment group	Delayed *n* = 33 Early *n* = 69	Survival Age-adjusted survival mRS
Eklund et al. [22]	2023	Sweden	2008–2011	Prospective cohort study	NPH patients *n* = 140 Matched controls *n* = 325	Overall survival
Pyykkö et al. [9]	2018	Finland	1993–2010	Retrospective cohort study	NPH patients *n* = 283 Matched controls *n* = 259	Overall survival
Klinge et al. [23]	2023	USA	2015–2021	Retrospective cohort study	NPH patients *n* = 208	Charlson comorbidity index Gait score Continence score mRS

**Table 2 neurolint-16-00107-t002:** Estimated hazard ratios (HRs) and corresponding 95% confidence intervals regarding progression-free survival in the one-stage meta-analysis continence score, gait score, and mRS.

**Gait Score**		
Study, year	Estimated HR	Estimated 95% CI
Andrén et al. 2020 [14]	2.23	1.76–2.81
Klinge et al. 2023 [23]	1.99	0.98–4.04
Entire IPD cohort (*n* = 1043)	2.25	1.81–2.81
**Continence Score**		
Study, year	Estimated HR	Estimated 95% CI
Andren et al. 2020 [14]	1.81	1.43–2.29
Klinge et al. 2023 [23]	1.13	0.60–2.13
Entire IPD cohort (*n* = 1022)	1.66	1.33–2.06
**mRS**		
Study, year	Estimated HR	Estimated 95% CI
Andren et al. 2020 [14]	2.24	1.74–2.90
Klinge et al. 2023 [23]	1.92	1.00–3.68
Entire IPD cohort (*n* = 956)	2.21	1.74–2.80

## Data Availability

The study includes all data. Primary data used for the analysis have already been published.

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
