# Peer review of "Survival After Shunt Therapy in Normal-Pressure Hydrocephalus: A Meta-Analysis of 1614 Patients"

_2035-8377, 2024, doi:10.3390/neurolint16060107_

Round 1
Reviewer 1 Report
Comments and Suggestions for Authors
I carefully read and reviewed the paper titled “Survival after Shunt Therapy in Normal Pressure Hydrocephalus: A meta-analysis of 1614 patients”. In present a meta-analysis, authors reviewed the impact of ventriculoperitoneal (VP) shunt therapy on survival in patients with normal pressure hydrocephalus (NPH), focusing on how baseline symptom burden, such as gait, continence, mRS scores, influences survival outcomes.
The authors defined the objective well enough; exploring survival time after VP shunt therapy for NPH and analyzing the role of baseline symptom burden. This is crucial since NPH treatment outcomes vary widely based on patient conditions.
Methodology of the meta-analysis is clear. The study uses individual patient data (IPD) from Kaplan-Meier charts reconstructed using statistical software, which improves the precision and reliability of survival estimates.
Authors reported impressive results. The results demonstrate that baseline symptoms (gait, continence, and mRS) significantly affect survival. For instance, patients with better gait had a mean survival of 8.24 years, while those with poor gait had 6.19 years. These differences are statistically significant and emphasize the importance of early intervention.
However, few issues should be revised:
- Structured abstract of the manuscript could be improved. The abstract mentions "two articles encompassing 1043, 1022, and 956 patients" which seems unclear, especially since only five articles were included in total. This wording could be restructured for clarity to avoid confusion about how many patients and articles contributed to the analyses.
- Authors stated that 216 records were initially screened, however, it would be helpful to include a brief mention of the criteria used to select the five studies. This could provide more context on the quality of evidence and ensure transparency regarding the eligibility process.
- Authors noted that HRs were presented for symptom stratifications and general population comparisons. Yet, the manuscript could benefit from a brief interpretation of what these hazard ratios mean in a clinical context. For instance, what does an HR of 2.25 for gait imply for patient management? Discuss please.
Author Response
Dear Reviewer
Thank you for reading our manuscript and critically reviewing it, which will help us improve it to a better scientific level and make it more understandable to the readership.
Comment 1
- Structured abstract of the manuscript could be improved. The abstract mentions "two articles encompassing 1043, 1022, and 956 patients" which seems unclear, especially since only five articles were included in total. This wording could be restructured for clarity to avoid confusion about how many patients and articles contributed to the analyses.
Response 1:
Thank you for your insightful feedback regarding the structured abstract. We recognize that the original phrasing might have led to confusion about the number of studies and patients contributing to the analyses. To address this, we have revised the sentence in the section “Abstract” for improved clarity as follows:
"In two of the five included studies, overall survival was stratified by gait score (1–4 vs. >4) in 1,043 patients, continence score (1–3 vs. >3) in 1,022 patients, and mRS (≤2 vs. >2) in 956 patients."
This revised wording makes it explicit that only two studies provided data for stratifying overall survival by these three variables, with distinct patient counts for each variable. The intention is to convey that although the overall analysis includes five studies, the stratifications by gait score, continence score, and mRS were conducted specifically in these two studies with the respective patient numbers. We hope this revision resolves any ambiguity and provides a clearer understanding of the data structure and analyses.
Comment 2:
- Authors stated that 216 records were initially screened, however, it would be helpful to include a brief mention of the criteria used to select the five studies. This could provide more context on the quality of evidence and ensure transparency regarding the eligibility process.
Response 2:
We initially screened 216 records. Our inclusion criteria focused on studies reporting long-term survival data in patients treated with ventriculoperitoneal (VP) shunts for normal pressure hydrocephalus (NPH). Specifically, studies were included if they provided Kaplan-Meier survival curves with corresponding number-at-risk tables. Studies were excluded if they were case reports, lacked survival data, or focused on other types of hydrocephalus. Following a two-step screening process involving independent reviewers, five studies met the eligibility criteria. These studies were assessed using the National Institutes of Health Quality Assessment Tool for Observational Cohort and Cross-Sectional Studies, ensuring a high standard of evidence for inclusion in our meta-analysis. However, only two studies in the literature provide longitudinal time-to-event data, enabling stratification by baseline symptom burden (gait score, continence score, and mRS). We believe pooling such longitudinal follow-up data is crucial to understanding how baseline symptom burden affects long-term survival and when clinical interventions might be most beneficial. A conventional meta-analysis pooling only dichotomized data (e.g., survival: yes or no) would be insufficient to address this question. However, this approach, as discouraged by the Cochrane Handbook [1], may not yield optimal conclusions. Time-to-event data are recognized as the most appropriate for survival analyses, and individual participant data (IPD) provide substantial advantages for such analyses [2]. The method with the R package IPDfromKM of Liu et al.'s [3] to reconstruct individual patient data from Kaplan-Meier curves has been utilized as a novel and validated method in our study. This enabled an IPD meta-analysis of time-to-event data, stratified by baseline symptom burden, to examine the long-term survival outcomes in patients undergoing VP shunt therapy for normal pressure hydrocephalus. Therefore, we have revised the section “Study selection and Study Characteristics” in the chapter Results
References
- Higgins JPT, Li T, Deeks JJ (editors). Chapter 6: Choosing effect measures and computing estimates of effect [last updated August 2023]. In: Higgins JPT, Thomas J, Chandler J, Cumpston M, Li T, Page MJ, Welch VA (editors). Cochrane Handbook for Systematic Reviews of Interventions version 6.5. Cochrane, 2024.
- Riley RD, Lambert PC, Abo-Zaid G. Meta-analysis of individual participant data: rationale, conduct, and reporting. BMJ. 2010 Feb 5;340:c221. doi: 10.1136/bmj.c221.
- Liu, N.; Zhou, Y.; Lee, J.J. IPDfromKM: reconstruct individual patient data from published Kaplan-Meier survival curves. BMC Med Res Methodol 2021, 21(1), 111. doi: 10.1186/s12874-021-01308-8.
Comment 3:
- Authors noted that HRs were presented for symptom stratifications and general population comparisons. Yet, the manuscript could benefit from a brief interpretation of what these hazard ratios mean in a clinical context. For instance, what does an HR of 2.25 for gait imply for patient management? Discuss please.
Response 3:
The reviewer is absolutely right that the clinical implications should be included in the discussion. Hence, we have revised the section “discussion” of the present manuscript. The hazard ratios (HRs) in this study provide valuable insights into how baseline symptom severity may impact survival in patients undergoing ventriculoperitoneal (VP) shunt therapy for normal pressure hydrocephalus (NPH). An HR of 2.25 for gait score (1–4 vs. >4) indicates that patients with more severe gait impairment face more than double the risk of mortality during follow-up compared to those with less severe impairment. This underscores the importance of early intervention for patients with moderate gait dysfunction, following a thorough diagnostic evaluation for NPH, to enhance survival outcomes. Similarly, HRs of 1.66 for continence score and 2.21 for mRS highlight the prognostic significance of these functional measures. These findings align with broader research linking physical performance metrics, such as gait speed, to survival outcomes in older adults. For instance, Studenski et al. [1] demonstrated in a pooled analysis of 34,485 individuals aged 65 years and older that gait speed is a robust predictor of life expectancy across diverse populations. Notably, urinary incontinence, a primary symptom of NPH, is independently linked to an elevated risk of mortality. In a study of 675 nursing home residents, Damian et al. [2] reported that incontinence was associated with a 24% increase in mortality risk, with severe cases correlating with a 44% higher risk.
References
- Studenski S, Perera S, Patel K, Rosano C, Faulkner K, Inzitari M, Brach J, Chandler J, Cawthon P, Connor EB, Nevitt M, Visser M, Kritchevsky S, Badinelli S, Harris T, Newman AB, Cauley J, Ferrucci L, Guralnik J. Gait speed and survival in older adults. JAMA. 2011 Jan 5;305(1):50-8.
- Damián J, Pastor-Barriuso R, García López FJ, de Pedro-Cuesta J. Urinary incontinence and mortality among older adults residing in care homes. J Adv Nurs. 2017 Mar;73(3):688-699.
Reviewer 2 Report
Comments and Suggestions for Authors
The disadvantage is that (1) the appearance of symbols such as "(1-3/>3)" in the abstract and the results of the third part is easy to cause misunderstanding and disagreement, and it is recommended to standardize the writing. (2) The full text lacks theoretical basis, including why the 95% confidence interval is taken instead of the 99.7% confidence interval.
Author Response
Dear Reviewer
Thank you for reading our manuscript and critically reviewing it, which will help us improve it to a better scientific level and make it more understandable to the readership.
Comment 1:
The disadvantage is that (1) the appearance of symbols such as "(1-3/>3)" in the abstract and the results of the third part is easy to cause misunderstanding and disagreement, and it is recommended to standardize the writing.
Response 1:
Thank you for your valuable feedback regarding the notation "(1-3/>3)" in the abstract and results sections. We understand that the use of such symbols may lead to potential misinterpretation, and we appreciate your recommendation to standardize the writing.
To address this, we have replaced "<3/>3" with "<3/≥4" throughout the manuscript body and the figures. This revised notation ensures greater clarity by explicitly defining the boundaries of the data categories. The use of "≥4" provides a clear and inclusive cutoff, minimizing ambiguity and enhancing the interpretability of our findings.
We have also ensured consistency in this notation across all relevant sections, including the abstract and results. We believe this adjustment will facilitate a smoother reading experience and foster alignment with standard scientific conventions.
Comment 2:
(2) The full text lacks theoretical basis, including why the 95% confidence interval is taken instead of the 99.7% confidence interval.
Response 2:
Thank you for your thoughtful feedback regarding the choice of the 95% confidence interval (CI) for hazard ratios in our analysis of longitudinal survival data. We greatly appreciate the opportunity to clarify our methodological approach.
The use of 95% confidence intervals in survival analysis, including univariate Cox regression, is standard practice in biomedical research. Here are several reasons supported by the literature:
- 95% Confidence Intervals Are Industry Standard:
- The 95% CI is a widely accepted standard in most statistical fields, including survival analysis, due to its balance between statistical power and precision. According to Altman et al. (1), 95% level provides an appropriate trade-off between confidence in the result and the precision of the estimate"​.
- Interpretability and Comparability:
- As noted by Schober and Vetter et al. (2), 95% CIs are used almost universally, providing a common framework that enhances comparability across studies​. In survival analysis, this comparability is crucial for meta-analyses and systematic reviews, like those pooling hazard ratios (HRs) across multiple studies​.
- Trade-Off with 99.7% Confidence Intervals:
- Increasing the CI to 99.7% results in wider intervals, reflecting greater uncertainty. While this reduces the risk of type I errors (false positives), it comes at the cost of reduced informativeness. Wider confidence intervals may obscure clinically relevant effects, reducing their practical utility (3)​.
- Standard Practice in Kaplan-Meier Survival Curves:
- Kaplan-Meier survival curves typically present 95% CIs for cumulative survival probabilities at different time points. This level is both statistically rigorous and visually interpretable for clinical decision-making​​.
- Regulatory and Clinical Relevance:
- Regulatory agencies and clinical guidelines, such as those by the FDA or EMEA, often expect 95% CIs in survival analyses. This standard allows for meaningful and accepted conclusions without overcomplicating the interpretation with excessively conservative intervals​.
References
- Altman DG, Gore SM, Gardner MJ, Pocock SJ. Statistical guidelines for contributors to medical journals. Br Med J (Clin Res Ed). 1983 May 7;286(6376):1489-93. doi: 10.1136/bmj.286.6376.1489.
- Schober, P., Vetter, T.R. (2018). Confidence intervals in clinical research. Anesthesia & Analgesia, 126(5), 1763-1768.
- Schober P, Bossers SM, Schwarte LA. Statistical Significance Versus Clinical Importance of Observed Effect Sizes: What Do P Values and Confidence Intervals Really Represent? Anesth Analg. 2018 Mar;126(3):1068-1072. doi: 10.1213/ANE.0000000000002798.